# "Colorectal Cancer Care Quality in a Developing Country: Insights from a Comparison of Teaching and Non-teaching Hospitals in Iran"

Mohammad Reza Rouhollahi[1], Mahdi Aghili[2], Saeed Nemati[1],
Mohammad Ali Mohagheghi[1], Farid Azmoudeh Ardalan[1], Habibollah Mahmoodzadeh[1],
Mehrzad Mirzania[1], Mohammad Shirkhoda[1], Seyed Hossein Yahyazadeh[3],
Ahad Muhammadnejad[4], Sepideh Abdi[1], Maedeh Zokaei Nikoo[5], Kazem Zendehdel[1]*

1 Cancer Research Center, Cancer Institute, Tehran University of Medical Sciences, Tehran, Iran,
2 Radiation Oncology Research Center, Cancer Institute, Tehran University of Medical Sciences, Tehran,
Iran, 3 Clinical Cancer Research Center, Milad Hospital, Iran University of Medical Sciences, Tehran, Iran,
4 Cancer Biology Research Center, Cancer Institute, Tehran University of Medical Sciences, Tehran, Iran,
5 Department of Outcome Research, Cleveland Clinic, Ohio, United States of America

* kzendeh@tums.ac.ir

## Abstract

### Background

Our study represents the first effort in the Eastern Mediterranean Region to identify disparities in the quality of colorectal cancer (CRC) care in Iran.

### Methods

We established a collaborative registry program for non-metastatic CRC patients to evaluate survival rates between teaching cancer centers (TCCs) and a high-volume, non-teaching, non-cancer center (NTNC). The study included a diverse patient population and considered various factors such as cancer stage, margin involvement, adherence to guidelines for adjuvant and neoadjuvant treatments, emergency surgeries, socioeconomic status, and risk of surgery. We utilized a multivariate Cox regression model and the targeted maximum likelihood estimator (TMLE) to analyze survival disparities in colorectal cancer between TCCs and the NTNC.

### Results

We recruited 668 CRC patients, including 320 with colon cancer and 298 with rectal cancer. Patients who underwent surgery at teaching cancer centers (TCCs) displayed significantly higher quality of care and better outcomes than those treated at the non-teaching, non-cancer center (NTNC). The adjusted hazard ratios (HR) were 1.97 (95% CI 1.21–3.21) for colon cancer and 1.54 (95% CI 1.01–2.55) for rectal cancer. Additionally, we observed significant causal mortality risk ratios (RR) based on hospital type for overall colorectal cancer (RR = 1.42, 95% CI 1.12–1.81) and specifically

**Data availability statement:** Since the patients did not consent to the public distribution of their data, and the Ethics Committee of Tehran University of Medical Sciences (TUMS) has not granted such rights to researchers, the data from this study cannot be published in public databases due to privacy concerns. The data may be available for research purposes upon reasonable request from the corresponding author (K.Z.) or the Chair of the Cancer Research Institute at TUMS. Requests can be directed via email to crc@tums.ac.ir. Please note that all new studies will require additional approval from the relevant Research Ethics Committee of TUMS in Iran.

**Funding:** This project has been conducted with a grant from the Cancer Institute of I.R. Iran, Sohrabi Cancer Charity, Grant No: 37369-202-01-97. Also, the collaborative registry program has been funded by the Research and Technology Deputy, Ministry of Health and Medical Education, Iran. Funders did not play any role in the study design, data collection and analysis, decision to publish or preparation of the manuscript.

**Competing interests:** The authors have declared that no competing interests exist.

for colon (RR = 1.48, 95% CI 1.04–2.11) and rectum cancer (RR = 1.39, 95% CI 1.01–2.02).

## Conclusion

The survival disparities in colon and rectal cancers between TCCs and NTNCs highlight a significant gap in CRC care in Iran. It is essential to expand this study nationally and implement the knowledge and experiences from TCCs in other hospitals to improve the quality of care and enhance patient outcomes.

## Introduction

Colorectal cancer (CRC) is recognized as the third most common cancer worldwide and ranks fourth in Iran [1,2]. Over the past few decades, treatment strategies for non-metastatic CRC have seen significant improvements, resulting in better patient outcomes [3]. However, these advances come against a backdrop of rising global cancer incidence and escalating treatment costs. For instance, projections indicate that between 2013 and 2035, CRC cases will increase by 59% in Australia and by 28% in the United States, while global mortality is expected to rise by 60% for colon cancer and 71% for rectal cancer. In this context, improving the quality of cancer care has become increasingly urgent [4].

The quality of cancer care in the late 20th century was defined as adherence to scientific evidence and clinical guidelines, and the assessment of its impact on disease outcomes [5]. With the publication of the National Comprehensive Cancer Network (NCCN) guidelines in 1996, systematic approaches have been improved to benchmark by measurement of the well-designed quality indicators and outcome in alignment of guidelines. Over time, many studies in developed countries focused on evaluating the outcomes of CRC surgery in high-volume or specialized cancer centers by comparing hospital mortality, 30-day postoperative mortality, and readmission rates due to surgical complications. Nevertheless, these studies were sometimes limited by incomplete clinical data, model deficiencies, and biases including self-interest and publication bias, leading to varied results across regions and inconsistent associations between hospital specialization, patient volume, and better surgical outcomes [6]. The results of these comparing studies were different in each of countries and regions and there was no necessarily significant relationship between better surgical outcomes and the specialization or high patient volume of hospitals [7–9]. In contrast colon cancer, in rectal cancer fewer studies were available, and almost all pointed to superior treatment outcomes in specialized and high-volume centers [10,11].

The subsequent development of clinical guidelines and the advent of evidence-based cancer care spurred further studies aimed at evaluating the effectiveness of these guidelines on patient outcomes. In this phase, cancer centers and high-volume hospitals voluntarily began recording detailed clinical data and establishing standardized clinical registries [12,13]. These registries enhanced the understanding of quality assessment methodologies and helped clarify disparities in cancer outcomes among

different centers. Moreover, the reported findings served as a valuable accreditation tool for healthcare providers and informed health system decision-makers [14].

Therefore through recent decade, to further develop the assessment of cancer care quality, a dynamic cycle consisting of the following components was proposed: Generating scientific evidence and developing clinical guidelines; translating evidence into practice; establishing an IT-based learning system for the collection of clinical data in cancer care practice; quality assessment using relevant indicators and valid data; enhancing quality and performance, and this cycle continues [15].

Advancements in survival study methodologies and increased access to high-quality data from standardized registries such as the National Cancer Database (NCDB) and the SEER program in the United States have enabled researchers to better quantify the net outcomes attributable to treatment at specific centers. These improvements underscore the importance of adopting a comprehensive conceptual model that accounts for all confounding factors; however, the model is sometimes justified depending on a country's unique conditions, data infrastructure, and research methods [16,17].

By now the majority of the cancer care quality assessment programs and related studies have been conducted in developed countries where there are well-established national cancer registries, such as NCDB, which contain patients and clinical factors as potential confounders for outcome evaluation [17]. Therefore, routine tracking of survival disparities is in some countries like the United Kingdom, aimed at assessing the quality of care on a national or subnational scale but not specifically at the hospital level [18,19]. In contrast, other countries, such as the U.S. [20], Japan [21,22], Switzerland [23], and Taiwan [24], have long employed outcome evaluations to compare hospitals across various criteria such as volume, specialization, or affiliation. In the United States, organizations such as the Agency for Healthcare Research and Quality (AHRQ) periodically report quality indicators—including the rate of resection of at least twelve lymph nodes in colon cancer surgery—across regional and national levels [25].

In low- and middle-income countries (LMICs), the establishment of cancer registries is becoming increasingly important for evaluating cancer care quality and survival [26,27]. Despite challenges in accessing essential clinical data in these regions, the concept of investigating potential disparities in care quality is emerging through collaborative research initiatives and conducting standardized registry-based studies. Nevertheless, only a few practical examples of such efforts exist to date [28]. In many LMICs, high-volume centers and comprehensive cancer centers have yet to implement robust registry systems, quality monitoring indicators, or evaluations of guideline adherence and their impact on patient outcomes. Gradually, these centers are receiving necessary training and are being required to adhere to clinical guidelines [29].

Surgery is a critical component in the multidisciplinary treatment of non-metastatic colorectal cancer. Successful surgical management requires not only expert surgeons but also the support of multidisciplinary teams and well-equipped hospitals that offer advanced radiologic investigations, intensive care units, and skilled nursing staff [30]. As previous studies addressed, in high-risk gastrointestinal surgeries, the outcome can be different or not based on the type of hospital, such as teaching status, localization (urban or rural), patient volume as well as cancer vs non-cancer hospitals [31,32]. Due to the relatively less complex nature of colon cancer surgeries compared with those for rectal tumors, many hospitals in both public and private sectors routinely perform colon surgeries, while rectal cancer cases are typically referred to teaching hospitals and specialized cancer centers [33].

In Iran, comprehensive cancer centers affiliated with major medical universities have been developing steadily over time. These centers have expanded their diagnostic and therapeutic capabilities for both common cancers, like CRC, and rare malignancies. They have also established multidisciplinary teams to address clinical complexities. Concurrently, high-volume non-teaching centers have been set up within the public sector and military services to provide routine cancer care. However, it is not always the case that all necessary modalities and specialties for comprehensive cancer care—along with fully integrated multidisciplinary decision-making teams—are present across these institutions. By the time the cohort study began (2013–2015), many cancer centers had increasingly prioritized adherence to standard clinical

guidelines, advanced their clinical data systems, and promoted clinical research. This evolution has prompted questions regarding whether teaching cancer centers, which benefit from academic expertise, demonstrate better process indicators and outcomes compared to high-volume non-cancer hospitals.

## Objectives

To develop an optimized model for assessing surgical quality in non-metastatic colorectal cancer—controlling for key confounders—to elucidate potential disparities between high-volume and teaching hospitals in developing countries

## Methods

### Collaborative quality registry program

A Collaborative Quality Registry Program has been established within the Quality Registry Network for Colorectal Cancer (QRN-CRC], which includes three comprehensive cancer centers and a high-volume non-teaching hospital located in the capital city of Tehran. This program encompasses various sub-projects to assess the quality of CRC care, all conducted voluntarily by network members.

Data were collected from a standardized clinical registry tool that is continuously developed and maintained by the research departments of each participating center. A steering committee, comprising representatives from the network members, was formed to facilitate this initiative. Additionally, a scientific advisory panel consisting of experts from various clinical disciplines and relevant research fields were established to develop the study methodology and implementation guidelines. This panel also played a crucial role in evaluating data, addressing ambiguities in complex decision-making scenarios, and ensuring the overall quality of the project.

### Hospitals

Four hospitals and comprehensive cancer centers participated in the QRN-CRC and contributed patient data for this study. The teaching cancer centers (TCCs] included one pioneer comprehensive cancer center and two teaching hospitals located in the Tehran metropolitan area, all serving as referral centers specializing in colorectal cancer (CRC) surgeries and related procedures. These TCCs allocate a significant portion of their surgical capacity to rectal cancer, which involves more complex surgical procedures, as well as the management of metastatic cancer cases. They also provide chemotherapy and radiation therapy.

Additionally, a large, referral non-teaching facility governed by the largest public health insurance organization participates in this study as the non-teaching non-cancer center (NTNC). Although the NTNC is not a university hospital, it handles a large number of patients, particularly those with non-metastatic CRC. While there is no universal threshold for defining high-volume hospitals for CRC surgery, this study considered hospitals performing approximately 100 non-metastatic CRC surgeries per year as high-volume centers. The high-volume center participating in the QRN-CRC network contributed 40% of the total patient data and 55% of the colon cancer cases in this study. To address the shortage of rectal cancer cases at the NTNC, eligible patients were enrolled from other non-teaching high-volume centers while being referred to the comprehensive cancer centers for adjuvant or neoadjuvant therapy in 2014.

### Patients

We included all non-metastatic (stages I-III) colorectal cancer patients (C18-20) with adenocarcinoma morphology (8140/3, 8480/3, 8490/3, 8020/3) undergoing curative surgery from early 2013 to late 2015 across QRN-CRC members and a few other high-volume hospitals. Metastasis status was determined through a comprehensive patient work-up, meeting guideline criteria and being free of tumor based on abdomen and pelvic computed tomography (CT) scans and a plain chest X-rays [34].

The exclusion criteria for this study included morphologies other than adenocarcinoma, tumor topography outside the colon or rectum (for example, anal tumors or cases where the tumor was reported in the rectosigmoid and could not be classified as either rectum or colon, i.e., merely not received neoadjuvant chemoradiotherapy was not considered as colon). Other exclusion factors involved patients without a determined pathological stage, and patients who had no National ID, e.g., immigrant patients, whose survival status could not be accurately tracked due to relocation.

## Database

To establish a clinical cancer registry database for the QRN-CRC, we customized the dataset based on registry standards of the U.S. National Cancer Database (NCBD). The detail of setting up this registry was published elsewhere [35]. Besides an experienced registrar was responsible for abstracting clinical data from patient medical records, three trained nurses who handled patient follow-up, data collection, a data analyst supported data management and analysis. The registry complies with patient privacy standards, ensuring researchers cannot access to personally identifiable information. Medical records, whether in paper format or scanned, from all patients who underwent surgery and had their final diagnoses coded using the specified topographic and morphologic codes were provided to the clinical registry unit by the medical records departments. Each patient file was required to include a surgical pathology report and cases of rectal cancer needed to undergo clinical staging. Cases diagnosed as stage IV at the time of diagnosis were excluded from the study. The hospital information management systems (HIS) did not contain the necessary clinical data, such as final diagnosis codes, and were used solely for patient follow-up (e.g., for the last patient visit) when phone contact with the patient was unsuccessful.

## Variables

This study was conducted based on a conceptual framework that includes input factors (both patient- and tumor-related), process factors (treatment interventions, excluding surgery, which is the primary focus of this study), and output indicators (such as five-year survival rates and early mortality rates). The input factors related to hospital status—such as bed availability, expert nursing staff, and necessary surgical infrastructure—were assumed to be similar between the two comparison groups (see Fig 1).

The primary independent variables in this study included potential prognostic factors that could influence patient survival. The first group comprises patient-related factors, including demographic variables such as age and gender and socioeconomic factors, represented by residential status (urban, suburban, or rural areas) and health insurance coverage. In Iran, citizens are enrolled in various health insurance plans based on their financial capacity, each offering different levels of coverage. Basic insurance plans provide minimal coverage, whereas supplementary insurance plans offer more comprehensive coverage, typically accessible to individuals with higher financial means. Some supplementary plans provide moderate coverage, ensuring a minimum level of protection beyond the basic plans.

Additionally, patient tolerance for surgery is a critical factor within the treatment process. This tolerance is primarily assessed preoperatively by anesthesiologists during the surgical risk assessment. An inaccurate evaluation of this risk could lead to adverse outcomes, such as decreased survival rates and increased post-surgical mortality. Consequently, this factor is considered both patient- and treatment-related. The surgical risk data were derived from evaluations documented by anesthesiologists, based on the preoperative assessment following the American Society of Anesthesiologists (ASA) scoring system [36].

Regarding tumor-related factors, we considered tumor location (i.e., colon or rectum), tumor morphology, tumor stage (i.e., stages I-III, excluding metastatic cases), and tumor grade. Tumor condition is particularly significant due to the potential for emergencies, such as obstruction or bleeding, which may arise from both the nature of the tumor and inadequate preoperative assessment.

| | Patient | Tumor |
|---|---|---|
| **INPUT** | • Gender<br>• Age<br>• Residential Status †<br>• Insurance Coverage †<br>• Risk of surgery | • Topography (Colon vs. Rectum)<br>• Morphology<br>• Tumor Stage<br>• Grade of tumor |
| **PROCESSES** | ***Neoadjuvant Chemoradiation Therapy ‡***<br>• Providing neoadjuvant chemoradiation therapy in adherence to the guideline<br><br>***Surgery and pre-and post-surgical assessment\****<br>• Assessment of risk of surgery (pre-surgical assessment by anesthesiologist)<br>• Emergent vs. Elective Surgery (pre-surgical assessment by surgeon)<br>• Margin Involvement by Tumoral Tissue (surgical procedure)<br>• Margin assessment (post-surgical assessment by pathologist)<br><br>***Adjuvant Chemotherapy***<br>• Administrating adjuvant chemotherapy in adherence to the guideline \*\* | |
| **OUTPUT** | • Five-year overall survival<br>• 30, 60, 90, 180-day mortality rate | |

\* These factors may be influenced by elements unrelated to the quality of surgical interventions (e.g., patient or
Tumor-related factors). Therefore, they have been included in the model.
\*\* For those who are at stage III and high-risk stage II patients in adherence to the guideline
† These are proxies of patients' socioeconomic status
‡ Only for rectal cancer

**Fig 1. Framework of factors affecting outcome in colorectal cancer in this study.**

We also considered treatment factors, including surgery, neoadjuvant chemoradiation for rectal cancer, and adjuvant chemotherapy to identify colorectal cancer (CRC) cases. We utilized the Clinical Practice Guidelines in Oncology (NCCN) for colon and rectal cancers, version 3.2014, published by the National Comprehensive Cancer Network [37,38]. According to these guidelines, the high-risk group for stage II colon and rectal cancers should receive adjuvant chemotherapy. Factors indicating a high risk of recurrence include poorly differentiated histology, lymphatic/vascular invasion, perineural invasion, bowel obstruction, localized perforation, fewer than twelve lymph nodes examined, and close, indeterminate, or involved margins. All the information required to identify the high-risk group within stage II cases was available in this registry.

The five-year survival rate served as the dependent variable in this study. Assessing disease-free survival complexities posed challenges, making it difficult to accurately inquire about recurrence or metastasis from patients or their family members during phone interviews. Mortality rates at 30-, 60-, 90-, and 180-days post-surgery were also compared between two groups.

## Follow-up

Trained nurses reviewed patient data in the hospital information systems (HIS) and medical records units. For patients who did not have subsequent visits, trained registrars conducted phone interviews with approximately 68% of patients, their families, or caregivers, completing a comprehensive checklist regarding post-surgery medical events, including chemotherapy, recurrence or distant metastasis, and mortality. For patients who could not be contacted for follow-up, their vital status was verified using their national identification number in the National Civil Registry database. The scientific advisory panel ensured clinical accuracy by reviewing documents and conducting follow-ups as necessary.

## Statistical analysis

Demographic, clinical, and treatment characteristics were summarized by hospital type. Unadjusted survival rates were assessed using Kaplan-Meier curves. A univariate analysis was conducted to identify potential factors influencing patient

survival. Multicollinearity testing was performed to explore correlations among significant independent factors identified in the univariate analysis before developing the regression model. The tumor grade and morphology were excluded from the multivariable models based on the multicollinearity matrix. The non-proportional hazard test indicated violations of the proportional hazards assumption for specific age groups in colon cancer. As a result, the final model incorporated the age variable in a time-varying format. We applied a stepwise approach to fit multivariate Cox regression models for overall colorectal cancer and for colon and rectum cancers separately. Analyses were performed using Stata software version 14 (StataCorp).

We utilized a doubly robust estimation method called the targeted maximum likelihood estimator (TMLE) to minimize potential selection bias. This method used R software and machine learning algorithms to estimate the mortality rate ratio between different types of hospitals, with teaching cancer centers (TCCs) as the reference group. In applying TMLE for colon and rectum cancer cases, we first predicted the outcome (i.e., mortality rate) based on various covariates using flexible machine learning algorithms, such as random forests and gradient boosting. Next, we estimated the propensity score, which represents the probability of receiving surgery at TCCs (the reference group) compared to non-teaching, non-cancer centers (NTNCs), using a machine learning model based on the same covariates. TMLE then adjusted the initial outcome model estimate to correct any residual bias, ensuring the final estimate aligned with the outcome and treatment models. Lastly, we estimated the treatment effect based on the targeted model, allowing for comparing surgery outcomes between TCCs and the NTNC. All statistical tests were conducted with two-sided P values, and statistical significance was set at 0.05.

### Ethics and other permissions

The study adhered to the ethical considerations based on the ethics committee approval of Tehran University of Medical Sciences (IR.TUMS.VCR.REC.1396.4202), and followed the Strengthening the Reporting of Observational Studies in Epidemiology (STROBE), reporting guideline designated for cohort studies. As per the approval from the ethics committee, obtaining consent from participants is not required for this study.

### Results

A total of 668 patients newly diagnosed with non-metastatic colorectal cancer between 2013 and the end of 2015 were enrolled in this study, including 370 (55.4%) colon and 298 (44.6%) rectum cancers. Fig 2 illustrates the patient inclusion from the TCCs and the NTNC. Overall, 404 (60%) patients underwent surgery at TCCs (41.5% colon cancer and 58.5% rectal cancer), while the remainder were registered at the NTNC (76.3% colon and 23.7% rectal cancers). The pathological stage was known for all cases, and the clinical stage was determined for 246 (82.6%) of the rectal cancer cases. The average follow-up time for patients was $3.03 \pm 1.8$ years. During this follow-up period, a total of 189 (28.3%) subjects died, of which 176 (93.1%) were attributed to colorectal cancer (CRC), and 13 cases (6.9%) resulted from competing risks.

A significant difference was observed between TCCs and the NTNC regarding the frequencies of colon and rectal cancers. Specifically, 45.1% of colon cancer patients and 74.8% of rectal cancer patients underwent surgery at TCCs ($p < 0.01$). In both TCCs and the NTNC, 44.9% of colon cancers were classified as locally advanced. However, for rectum cancers, 60.1% of patients in TCCs and 52.4% in the NTNC were categorized as locally advanced ($p < 0.01$), indicating that TCCs performed more advanced rectal surgeries than the NTNC. The proportion of high-risk patients was higher in the NTNC (22.6%) than in TCCs (18.2%) (p-value 0.04). Likewise, the NTNC had significantly more urgent surgeries (11.3%) compared to TCCs (5.7%) (p-value $< 0.01$) (Table 1). Additionally, statistically significant differences were noted between the NTNC and TCCs in the year of surgery, insurance coverage, and tumor grade ($p < 0.01$) and in adherence to guidelines for adjuvant chemotherapy ($p = 0.01$) (see Table 1).

Unadjusted survival rates in TCCs vs NTNC for colorectal, colon and rectum are depicted separately by Kaplan-Meier diagram in Fig 3.

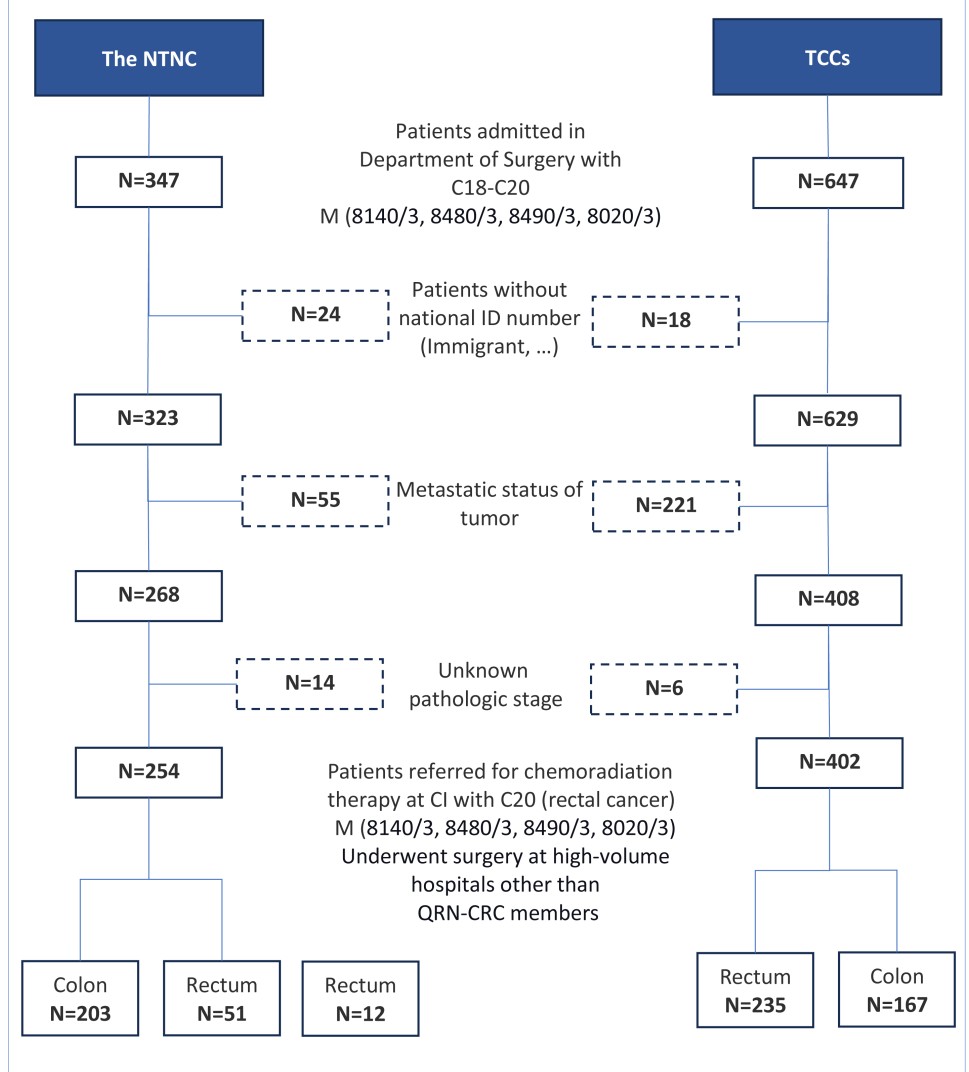

**Fig 2. Patient flow diagram and number of patients recruited from the teaching cancer centers (TCCs) and a high-volume non-teaching non-cancer center (NTNC) to study the CRC care quality in Iran.**

The multivariable Cox regression analysis revealed that patients who underwent surgery at the NTNC had significantly lower survival rates compared to those treated at TCCs for overall colorectal cancer (HR = 1.49, 95% CI 1.09–2.04), colon (HR = 1.97, 95% CI 1.21–3.21) and rectum cancers (HR = 1.54, 95% CI 1.01–2.55). We observed a higher risk of death for colorectal and colon cancers in patients living in suburban areas compared to those in rural areas [colorectal: HR = 1.80, 95% CI 1.13–2.85; colon: HR = 2.21, 95% CI 1.14–3.94]. Furthermore, patients with involvement of the surgical margin had a higher risk of death compared to those with margin-free resections [colorectal: HR = 2.68, 95% CI 1.70–4.22; colon: HR = 2.64, 95% CI 1.46–4.79]. Diagnosis at stage III was associated with a greater risk of death compared to those with earlier-stage tumors [colorectal: HR = 2.72, 95% CI 1.89–3.91; colon: HR = 3.20, 95% CI 1.99–5.13]. In addition, non-adherence to the adjuvant chemotherapy guidelines was linked to higher mortality rates than adherence [colorectal: HR = 1.71, 95% CI 1.21–2.59; colon: HR = 2.38, 95% CI 1.32–4.28]. For rectum

**Table 1. Demographic, Tumor, and Treatment Characteristics in Two Types of Hospitals (TCCs vs NTNCs) in colorectal cancer.**

| Variables | Cancer Centers (TCCs) n (%) | Non-Cancer Centers (NTNCs) n (%) |
|---|---|---|
| **Total** | 402 (60.18) | 266 (39.82) |
| **Gender** | | |
| Male | 237 (59.0) | 173 (65.0) |
| Female | 165 (41.0) | 93 (35.0) |
| **Age Categories** (Years) | | |
| <50 | 113 (28.1) | 59 (22.2) |
| ≥50 & <70 | 207 (51.5) | 146 (54.9) |
| ≥70 | 82 (20.4) | 61 (22.9) |
| **Year of Surgery** | | |
| 2013 | 79 (19.6) | 80 (30.1) |
| 2014 | 124 (30.9) | 100 (37.6) |
| 2015 | 199 (49.5) | 86 (32.3) |
| **Residential Status** | | |
| Urban | 335 (83.3) | 209 (78.6) |
| Suburban | 27 (6.7) | 23 (8.6) |
| Rural | 40 (10.0) | 34 (12.8) |
| **Insurance Coverage** | | |
| High | 58 (14.5) | 52 (19.6) |
| Average | 219 (64.4) | 202 (75.9) |
| Low | 60 (14.9) | 10 (3.8) |
| No Insurance | 25 (6.2) | 2 (0.7) |
| **Topography** | | |
| Colon | 167 (41.5) | 203 (76.3) |
| Rectum | 235 (58.5) | 63 (23.7) |
| **Morphology** | | |
| Adenocarcinoma; NOS | 361 (89.8) | 251 (94.4) |
| Mucinous A. | 37 (9.2) | 9 (3.4) |
| Signet Ring Cells A. | 2 (0.5) | 6 (2.3) |
| Undiff. Carcinoma | 2 (0.5) | 0 |
| **Margin Involvement** | | |
| Free | 359 (89.3) | 242 (91.0) |
| Closed | 9 (2.2) | 4 (1.5) |
| Involved | 34 (8.46) | 20 (7.5) |
| **Stage at diagnosis** | | |
| Stage I | 29 (7.2) | 19 (7.14) |
| Stage II | 120 (29.8) | 106 (39.8) |
| Stage III | 218 (54.2) | 124 (46.6) |
| Unknown | 35 (8.7) | 17 (6.4) |
| **Grade** [a] | | |
| Well-differentiated | 108 (26.9) | 113 (42.5) |
| Moderately diff. | 198 (49.2) | 124 (46.7) |
| Poorly diff. | 36 (9.0) | 28 (10.5) |
| Not applicable [b] | 55 (13.9) | 1 (0.4) |

*(Continued)*

**Table 1.** (Continued)

| Variables | Cancer Centers (TCCs) n (%) | Non-Cancer Centers (NTNCs) n (%) |
|---|---|---|
| **Emergent or Elective** | | |
| Elective surgery | 379 (94.3) | 236 (88.7) |
| Emergent Surgery | 23 (5.7) | 30 (11.3) |
| **Risk of surgery** | | |
| Mild | 117 (29.1) | 55 (20.7) |
| Moderate | 212 (52.7) | 151 (56.8) |
| Sever | 73 (18.1) | 60 (22.5) |
| **Adherence to the Guideline in Adjuvant chemotherapy** | | |
| Adherent | 246 (61.2) | 178 (66.9) |
| Non-adherent | 77 (19.1) | 49 (18.4) |
| Not applicable [c] | 17 (4.2) | 18 (6.8) |
| Unknown | 62 (15.4) | 21 (7.9) |

[a]Total percentage for this variable will not yield 100%, because unknown categories are not shown.

[b]This category is related to the pathologic complete response (pCR) to the neoadjuvant chemoradiation therapy in some rectal cancer cases.

[c]In this category, patients died before starting or had low performance for adjuvant chemotherapy.

[**]The table describing the variability of characteristics does not require p-values, in accordance with the STROBE guidelines [39].

cancer, patients with involved margins (HR = 3.98, 95% CI 2.00–7.90) and those diagnosed at a more advanced stage (HR = 2.60, 95% CI 1.40–5.03) also showed a higher risk of death compared to their corresponding reference groups (Table 2).

## Discussion

This study found that patients with colon or rectal cancer who underwent surgery at TCCs had a better prognosis compared to those treated at NTNCs. We confirmed that being in a locally advanced stage and having residual malignant tissue in the surgical margin were significant prognostic factors for both colon and rectal cancers. Additionally, for colon cancer, non-adherence to adjuvant chemotherapy guidelines was a prognostic factor. Improving these two factors can be considered by NTNCs to enhance patient survival rates. Furthermore, residing in suburban areas was associated with poorer outcomes for colon cancer patients. Contrary to our expectations, factors such as the need for urgent surgery and patients with a high surgical risk did not correlate with survival outcomes in colorectal cancer. This is the first study in the Eastern Mediterranean Region to compare survival disparities in colorectal cancer between cancer centers and non-teaching hospitals.

We developed a robust model encompassing four categories of factors: patient-related factors such as age, gender, socioeconomic status, and surgical risk; tumor-related factors including stage, margin status, and urgency of surgery; neo-adjuvant treatment (specific to rectal cancer); and adherence to adjuvant treatment guidelines. Additionally, we included surgery and related factors as a fourth category without a further breakdown into specific elements. Our findings indicate that the type of hospital where patients undergo surgery for primary tumors, both colon and rectal cancers, may significantly impacts survival outcomes.

In addition to achieving a clear tumor margin and ensuring the administration of adjuvant chemotherapy for eligible patients, a separate study with an appropriate methodology and model design could be conducted to determine the proportion attributable to adherence quality indicators in the treatment process and the survival differences between these two hospital groups. Recognized indicators, such as adequate lymph node dissection (a minimum of 12 lymph nodes in colon resection surgery)

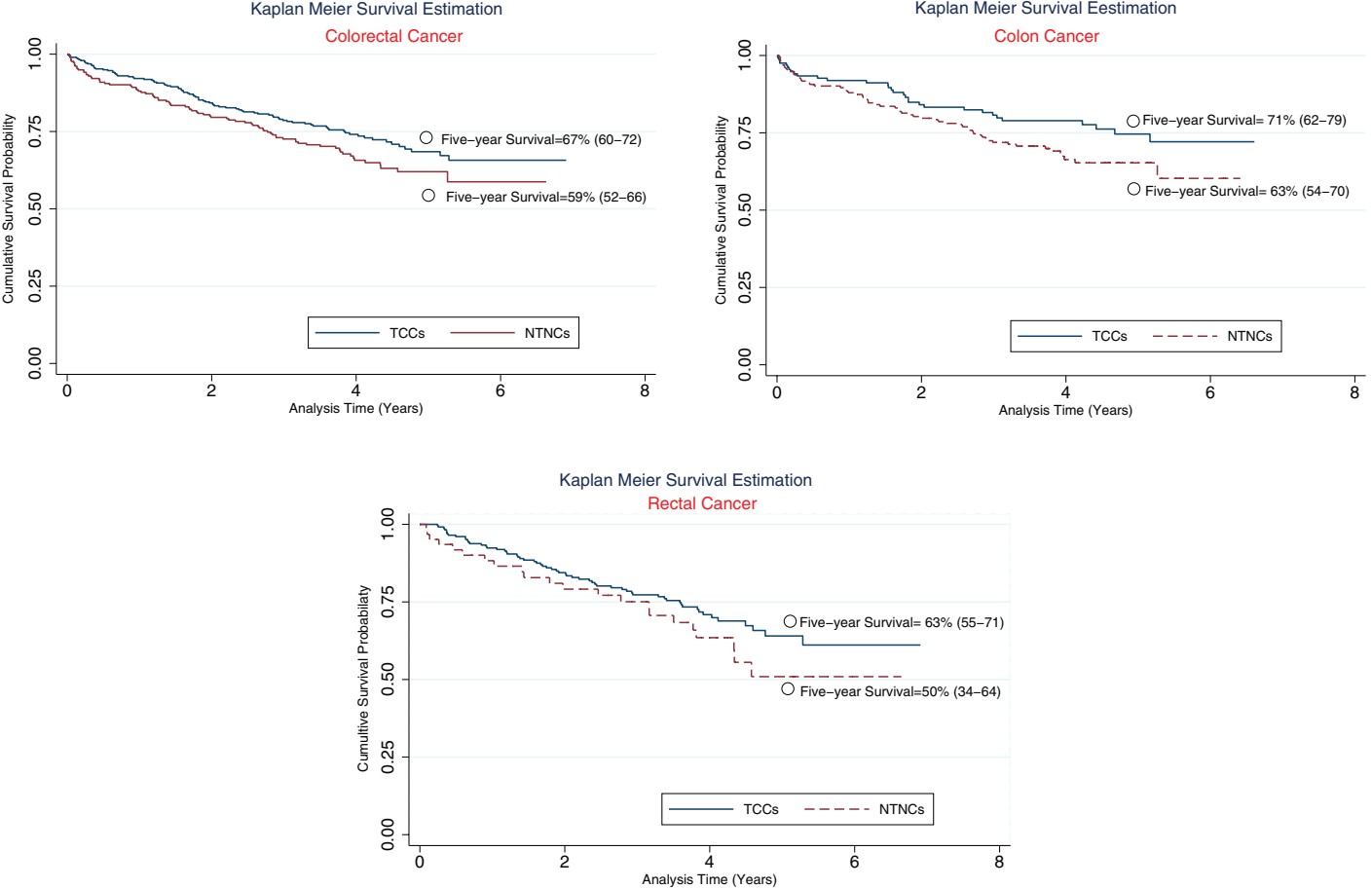

**Fig 3. Unadjusted Survival Rate in Colorectal. (a), Colon (b), and Rectum (c) Cases in Each Type of Center.**

and adherence to the recommended interval between the completion of neoadjuvant therapy and surgery (at least 12 weeks for rectal cancer), would be incorporated into this model. This would encourage non-academic centers involved in this study to improve compliance with quality indicators; however, this aspect falls beyond the scope of the present study.

There are some studies reported disparities in patient outcomes related to hospital volume, reputation, ranking, type of hospital (teaching vs. non-teaching), and specialty level (cancer vs. non-cancer centers) in high-income countries [16,40,41] In 2019, only 27% of colon cancer patients in the U.S. underwent surgery at high-volume urban hospitals, which are associated with better outcomes. However, this association was not statistically significant when considering the teaching status of the hospitals [31]. In another study on rectal cancer, surgery performed at higher-volume teaching and comprehensive cancer centers improved long-term survival for U.S. patients between 1998 and 2007 [33]. Conversely, in the Southern Netherlands, data from 2008 to 2011 showed no significant differences in short-term survival rates for CRC surgery when comparing the lowest and highest-volume hospitals [42]. However, no studies have focused on this issue in low-income countries.

In our study, the subgroup analysis for locally advanced tumors (stage III) indicated that survival probabilities for all colorectal and colon cancers were significantly higher in TCCs compared to the NTNC. However, we did not have sufficient statistical power to demonstrate a significant difference in survival rates between hospital types for rectal cancer.

**Table 2. Muti-variable Survival Analysis, Cox model, in colorectal, colon and rectum.**

| Variables | Total Colorectal N = 668 | Colon N = 370 | Rectum N = 298 |
|---|---|---|---|
| **Cancer vs Non-cancer Centers** | Adjusted HR (CI95%) | Adjusted HR (CI95%) | Adjusted HR (CI95%) |
| Cancer Centers | Ref. | Ref. | Ref. |
| Non-Cancer Centers | **1.49 (1.09-2.04)** | **1.97 (1.21-3.21)** | **1.54 (1.01-2.55)** |
| **Gender** | | | |
| Male | Ref. | Ref. | Ref. |
| Female | 0.96 (0.70-1.32) | 0.99 (0.63-1.53) | 0.90 (0.54-1.47) |
| **Age** | | | |
| Age < 45 | Ref. | Ref. | Ref. |
| 45 ≥ Age < 55 | 0.74 (0.42-1.29) | 0.51 (0.25-1.05) | 0.97 (0.44-2.14) |
| 55 ≥ Age < 65 | 0.56 (0.29-1.10) | 0.59 (0.30-1.14) | 0.62 (0.23-1.67) |
| 65 ≥ Age < 75 | 0.66 (0.33-1.33) | 0.55 (0.28-1.10) | 0.71 (0.25-2.04) |
| Age ≥ 75 | 0.91 (0.41-2.03) | 0.65 (031-1.36) | 1.87 (0.59-5.94) |
| **Residential status*** | | | |
| Urban | Ref. | Ref. | – |
| Marginal urban | **1.80 (1.13-2.85)** | **2.21 (1.14-3.94)** | – |
| Rural | 0.89 (0.55-1.46) | 0.93 (0.48-1.79) | – |
| **Margin of Surgery** | | | |
| Free | Ref. | Ref. | Ref |
| Involved | **2.68 (1.70-4.22)** | **2.64 (1.46-4.79)** | **3.98 (2.00-7.97)** |
| **Stage at diagnosis** | | | |
| Early Stage (I-II) | Ref. | Ref. | Ref |
| Locally Advanced (III) | **2.72 (1.89-3.91)** | **3.20 (1.99-5.13)** | **2.61 (1.36-5.03)** |
| Unknown | **1.91 (1.02-3.57)** | – | **2.81 (1.28-6.16)** |
| **Emergent or Elective ****  | | | |
| Elective surgery | Ref. | – | Ref |
| Emergent Surgery | 1.38 (0.82-2.30) | – | 1.57 (0.71- 3.48) |
| ****Risk of surgery** | | | |
| Mild | Ref. | – | |
| Moderate | 1.12 (0.64-1.96) | – | 1.09 (0.49-2.41) |
| Sever | 1.19 (0.59-2.39) | – | 1.60 (0.61- 4.18) |
| ***Adherence to the Guideline in Adjuvant chemotherapy** | | | |
| Adherent | Ref. | Ref. | – |
| Non-adherent | **1.77 (1.21-2.59)** | **2.38 (1.32-4.28)** | – |
| Unknown | **1.81 (1.01-3.01)** | 1.67 (0.68-4.10) | – |
| | | | |

*This variable could not be in the multiple regression model in the rectal cases because it did not have significant effect in univariable analysis.*

**This factor was kept out of multiple regression model in the colon cases because of its non-significant effect in univariable analysis.*

The TMLE approach revealed that the NTNC had a significantly higher mortality rate ratio (RR) compared to TCCs for overall colorectal cancer (RR = 1.42, 95% CI 1.12–1.81), and for colon cancer (RR = 1.48, 95% CI 1.04–2.11) and rectal cancer (RR = 1.39, 95% CI 1.01–2.02).

By calculating the 30-, 60-, and 90-day post-surgery mortality rates, under the assumption that early mortality was due to surgical complications and inadequate preoperative risk assessment. Our analysis revealed no significant difference in these mortality rates between the two comparison groups. However, the 90-day mortality for rectal cancer was lower

in TCCs compared to the NTNCs, albeit it was not statistically significant. For the 180-day post-surgery mortality, which serves as an indicator of incorrect tumor staging at diagnosis and may suggest that the tumor was likely in a metastatic state, there was no significant difference in mortality rates for colon and rectal cancer between TCCs and NTNCs (S1 Table).

Research on socioeconomic status (SES) has consistently demonstrated that cancer patients from disadvantaged backgrounds experience lower survival rates and higher mortality [43–45]. In our study, we found that living in a suburban area was an adverse prognostic factor for patients with colon cancer and CRC overall. However, we chose to exclude this factor from the multivariable model for rectal cases due to its lack of significance in the univariable analysis, likely attributed to the limited power of this study. Previous studies have examined the rural-urban divide in CRC survival, primarily relying on population-based cancer registries that emphasize the critical role of access to cancer care. In contrast, our study identified suburban residency as a marker of social marginalization and deprivation, linking it to poorer cancer survival outcomes within a hospital-based cancer registry framework. Future studies with larger sample sizes are necessary to validate our findings [46].

Although pathologic complete response (pCR) to neoadjuvant chemoradiation therapy has been associated with improved survival rates in rectal cancer [47], it showed a strong correlation with the type of hospitals in our study. Given that it may be considered an intermittent factor, we decided to exclude it from the multivariable analysis.

This study utilized the targeted maximum likelihood estimation (TMLE) approach, a novel matching method suited for cohort-based studies. While TMLE is considered more appropriate, other matching techniques, such as propensity score (PS) analysis, have also been applied in similar research [16]. Given that TMLE provides a more flexible and efficient approach, particularly in complex settings [48]. Our findings with TMLE confirmed the cox model results, particularly in rectal cancer, despite the relatively low sample size in this group.

A key advantage of this research is its structure, developed by self-motivated participants in a collaborative program. A small, trained registry team, overseen by a multidisciplinary advisory panel and a multicenter steering committee, maintained a standardized colorectal cancer registry and validated the study's findings. The robust survival analysis model also effectively controlled for confounding variables, ensuring an optimal sample size.

To evaluate whether loss to follow-up occurred randomly and to address potential selection bias, we compared the basic characteristics of the censored and non-censored groups, finding no significant differences (S2 Table).

International efforts have shown that providing measurable tools to support quality improvement initiatives can motivate clinical practitioners and healthcare providers [49]. Establishing registries, effective clinical data management, and monitoring the quality-of-care indicators and patient outcomes are now regarded as essential components of comprehensive cancer centers [29]. In this registry-based study, we included all eligible patients while avoiding the selection of specific individuals. Despite the small sample size, particularly for rectal cancer cases in the NTNC hospital, we observed a significant difference in patient survival. However, there were some limitations in this study. The loss to follow-up rate (see the S2 Table), missing stage-at-diagnosis data in a few rectal cancer cases, and absent variables such as additional socioeconomic status (SES) proxies, including patients' or their families' income, should be addressed in future research. Although this study aimed to control all confounding factors, to ensure the study's practicality, the registry was designed to include only the minimum necessary dataset. Furthermore, our focus on high-volume cancer and non-cancer centers in the capital city of Tehran may not fully represent the overall colorectal cancer patient population in Iran. Data from a larger network of TCCs and NTNCs is needed to represent the situation of CRC care quality in Iran.

In healthcare quality assessment methodology, elements like structural characteristics and resource allocation are classified as inputs. Evaluating healthcare processes through quality indicators and outcome analysis can help identify disparities, which may then warrant a deeper investigation into inputs to uncover the root causes of these differences [50]. Once we identify any outcome disparities while controlling for potential confounders, we can plan to further investigate the surgical department by examining its specific quality indicators. For instance, approved indicators related to the surgical

process include achieving a dissection of at least 12 lymph nodes during colon cancer surgery and ensuring an interval of 8–12 weeks from the end of neoadjuvant chemoradiation therapy to the rectal cancer surgery. Additionally, comparing hospital infrastructures may help uncover the root causes of these differences.

According to the definition provided in the manuscript, the primary component of quality evaluated in this study is "effectiveness," while other components—such as "acceptability," "equity," "optimality," and "legitimacy"—are undoubtedly important but fall outside the scope of our study.

## Conclusions

Our study identified significant disparities in the quality of care for non-metastatic CRC between TCCs and NTNCs. The TCCs exhibited better outcomes, likely due to their academic and research-focused environments, collaboration within multidisciplinary teams, and commitment to scientific endeavors, which enhance the care provided to CRC patients. In contrast, there remains considerable room for improved CRC survival rates at NTNCs. These findings offer a valuable opportunity for health policymakers and authorities to leverage the educational and scientific frameworks established in TCCs. By providing support and resources to non-cancer facilities, it may be possible to improve the overall quality of cancer care across various healthcare settings. We recommend expanding the scope of this project to include other high-volume hospitals in different provinces of Iran or even in the Eastern Mediterranean region to provide a more accurate picture of the quality of CRC care in this area.

## Supporting information

**S1 Table. Comparing the censored and non-censored patients according to different patient-related variables among colorectal cancer patients in Iran.**
(DOCX)

**S2 Table. Short-term mortality rate in TCCs vs NTNCs in colon and rectal cases in Iran.**
(DOCX)

## Acknowledgments

The authors thank WE4H Writing and Editing Company in B.C., Canada, for the English editing of the manuscript.

## Author contributions

**Conceptualization:** Mohammad Reza Rouhollahi, Saeed Nemati, Mohammad Shirkhoda, Seyed Hossein Yahyazadeh, Ahad Muhammadnejad, Maedeh Zokaei Nikoo, Kazem Zendehdel.

**Data curation:** Mohammad Reza Rouhollahi, Saeed Nemati, Farid Azmoudeh Ardalan, Mohammad Shirkhoda, Seyed Hossein Yahyazadeh, Ahad Muhammadnejad, Sepideh Abdi, Maedeh Zokaei Nikoo.

**Formal analysis:** Mohammad Reza Rouhollahi, Saeed Nemati, Kazem Zendehdel.

**Funding acquisition:** Mohammad Ali Mohagheghi, Kazem Zendehdel.

**Investigation:** Mohammad Reza Rouhollahi, Ahad Muhammadnejad, Sepideh Abdi, Maedeh Zokaei Nikoo.

**Methodology:** Mohammad Reza Rouhollahi, Mahdi Aghili, Saeed Nemati, Mohammad Ali Mohagheghi, Farid Azmoudeh Ardalan, Habibollah Mahmoodzadeh, Mehrzad Mirzania, Mohammad Shirkhoda, Seyed Hossein Yahyazadeh, Ahad Muhammadnejad, Maedeh Zokaei Nikoo, Kazem Zendehdel.

**Project administration:** Mohammad Reza Rouhollahi, Kazem Zendehdel.

**Resources:** Mohammad Ali Mohagheghi, Kazem Zendehdel.

**Software:** Mohammad Reza Rouhollahi, Saeed Nemati, Sepideh Abdi, Maedeh Zokaei Nikoo, Kazem Zendehdel.

**Supervision:** Mahdi Aghili, Mohammad Ali Mohagheghi, Farid Azmoudeh Ardalan, Habibollah Mahmoodzadeh, Mehrzad Mirzania, Mohammad Shirkhoda, Seyed Hossein Yahyazadeh, Kazem Zendehdel.

**Validation:** Mahdi Aghili, Mohammad Ali Mohagheghi, Farid Azmoudeh Ardalan, Habibollah Mahmoodzadeh, Mehrzad Mirzania, Mohammad Shirkhoda, Ahad Muhammadnejad.

**Visualization:** Mohammad Reza Rouhollahi, Saeed Nemati, Sepideh Abdi, Kazem Zendehdel.

**Writing – original draft:** Mohammad Reza Rouhollahi.

**Writing – review & editing:** Mahdi Aghili, Mohammad Ali Mohagheghi, Farid Azmoudeh Ardalan, Habibollah Mahmoodzadeh, Mehrzad Mirzania, Mohammad Shirkhoda, Seyed Hossein Yahyazadeh, Ahad Muhammadnejad, Kazem Zendehdel.

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
