## [Decision Letter · Decision Letter 0]

29 Jul 2024

PONE-D-23-43777Bridging gaps in colorectal cancer care quality in developing countries: Navigating disparities with the right approach, analytical methods, and minimum essential dataPLOS ONE

Dear Dr. Zendehdel,

Thank you for submitting your manuscript to PLOS ONE. After careful consideration, we feel that it has merit but does not fully meet PLOS ONE’s publication criteria as it currently stands. Therefore, we invite you to submit a revised version of the manuscript that addresses the points raised during the review process. Esteemed reviewers have raised major issues regarding the study objectives and methodology which need appropriate revisions and replies to the queries to enable further evaluation of this submission. Please follow the received comments to revise the manuscript and material.

We look forward to receiving your revised manuscript.

Kind regards,

Sina Azadnajafabad, MD, MPH

Academic Editor

PLOS ONE

2. In the online submission form, you indicated that [The data for this study are available on reasonable request from the corresponding author (K.Z.).]. 

Additional Editor Comments (if provided):

Reviewers' comments:

Reviewer's Responses to Questions

**Comments to the Author**

1. Is the manuscript technically sound, and do the data support the conclusions?

Reviewer #1: Yes

Reviewer #2: Yes

Reviewer #3: Partly

Reviewer #4: Partly

2. Has the statistical analysis been performed appropriately and rigorously? 

Reviewer #1: Yes

Reviewer #2: Yes

Reviewer #3: Yes

Reviewer #4: Yes

3. Have the authors made all data underlying the findings in their manuscript fully available?

Reviewer #1: Yes

Reviewer #2: No

Reviewer #3: No

Reviewer #4: No

4. Is the manuscript presented in an intelligible fashion and written in standard English?

Reviewer #1: Yes

Reviewer #2: No

Reviewer #3: No

Reviewer #4: Yes

5. Review Comments to the Author

Reviewer #1: The study utilizes a robust methodology to assess survival disparities in non-metastatic colorectal cancer patients across different types of hospitals in Iran. The focus on high-volume teaching versus non-teaching hospitals provides valuable insights into the impact of hospital type on patient outcomes.

Comments:

1. Some parts of the manuscript are a bit confusing and could be better structured. For instance, the introduction should give a clearer and shorter summary of the research problem and goals via disparities.

2. The manuscript mentions the importance of the study, but it would be helpful to have a more in-depth discussion of other studies about colorectal cancer care differences in poorer countries. This would provide a stronger foundation for the research.

3. It's great that you included different types of hospitals, but it would be good to explain in more detail why you chose these specific hospitals and how they represent the overall healthcare system.

4. The information about the variables is thorough, but it would be useful to clarify exactly how you measured each variable and gathered the data. This would make it easier for others to repeat your study.

5. The TMLE method is complex, so it would be helpful to explain it more clearly.

6. Please explain why you chose the number of participants you did and if you calculated how many you needed to get reliable results (power analysis).

7. The discussion does a good job of putting the results in context, but it could go deeper into why the differences in care exist.

8. Based on your findings, please suggest specific actions or changes that could be made to improve the situation (or in further studies).

Reviewer #2: In the present study, the authors evaluated the outcomes and prognosis of colorectal, colon, and rectum cancers by the level of care provided in teaching and non-teaching non-cancer hospitals, using a data registry in Iran. The study subject is interesting. However, the use of language should be improved. Additionally, there are further comments and suggestions provided below:

**General comments**

1. There are some typos and grammatical issues within the text. Please proofread the entire manuscript.

2. Define abbreviations upon their first use in the text.

3. Use p<0.01 instead of p=0.00.

**Abstract**

1. Please clarify and elaborate more on the background and objective of the study.

2. Check the number: RR=1.39, 95% CI: 1.01-2.022. Should it be 2.02 or 2.022? Revise it in the text if necessary.

3. Revise the keywords based on MeSH.

**Introduction**

1. The SEER program in the US can also be discussed.

2. In the second paragraph, please elaborate more and mention some of the findings from countries that have subnational or hospital-level data.

**Methods**

1. Provide the exclusion criteria under the "Patients" subheading.

2. Cite the relevant references for the guidelines used in the study (Variables subheading).

3. It is recommended to provide an "Outcome" subheading and define the outcomes of interest for this study.

**Results and tables**

1. Table 1: Define what you mean by high, average, and low insurance coverage, and how you determine the risk of surgery. Also, it might be helpful to provide a column to represent the p-value between the groups for each variable.

2. You mentioned that comorbidities were also considered. Please provide the baseline characteristics for comorbidities in Table 1 and clarify which comorbidities were considered.

Reviewer #3: Thank you for the opportunity to review your important work examining the quality of colorectal cancer care. This study addresses a critical gap in understanding the disparities in care quality across different hospital types in the Eastern Mediterranean region. Below are my general comments and specific recommendations:

General comments:

1. The objective of the study, whether it focuses on the quality of care or prognosis, needs to be stated clearly at the outset.

2. The rationale and importance of this study, as well as its contribution to the body of knowledge on colorectal cancer care quality, need to be more clearly articulated.

3. Ensure that the outcome measures and study setting are explicitly defined.

4. Maintain consistent language and style throughout the manuscript.

5. Improve the writing style by starting with a stem statement followed by key points relevant to the study's objectives and direction.

Specific sections:

Abstract: Please revise the abstract to reflect the recommendations provided for the entire article.

Introduction

1. Include a section that describes the burden of colorectal cancer and the significance of monitoring and evaluating care outcomes.

2. Justify the selection of input, process, and outcome indicators commonly used to monitor care quality, enabling readers to understand the global and regional context. This will help readers understand the international and regional context of the study.

3. Follow statements such as "registry-based study has been advanced by a collaborative network" with an explanation of the network's function and contribution to improving care quality.

4. Describe the multidisciplinary team involved, the characteristics of well-equipped hospitals, and how these factors contribute to care quality.

5. Provide information on existing patient outcomes, survival rates, and process indicators used in colorectal cancer surgery. Discuss the advantages and disadvantages of these indicators.

6. Include evidence that patient outcomes for colorectal cancer vary based on hospital type (teaching status, cancer specialization) and location.

7. Clearly describe the infrastructure gaps relevant to colorectal cancer care quality that the study aims to highlight.

8. Refine the objective statement using the SMART criteria (Specific, Measurable, Achievable, Relevant, Time-bound).

Methods

1. Provide an overview of the healthcare systems in Iran, particularly colorectal cancer services, and any changes in service delivery over time.

2. Organize the Methods section to provide a clear contextual understanding of colorectal cancer treatment in Iran, followed by the role of different hospitals, the quality-network, and data collection processes. Conclude with detailed data management and analysis descriptions.

3. Clarify whether QRN-CRC is limited to specific hospitals or is a nationwide program. Provide the rationale for selecting the quality registry network and differentiate between hospital types.

4. Explain how clinical registry data is collated, whether through hospital information systems or retrospective patient tracking and case note reviews.

5. Specify the number of hospitals included in the study, the definition of high-volume hospitals, and the quality of data in the cancer registry.

6. Summarize variables in a table with clear definitions to aid reader comprehension. Include guidelines for patient treatment based on cancer stages as appendices, if necessary.

7. Clearly define the study's outcome or dependent variables.

8. Clarify the study duration and follow-up period. Provide details on patient follow-up timelines to help readers understand the duration of follow-up.

9. Describe potential selection bias reasons and the steps taken to mitigate it.

Results

1. Summarize the patient flow in a diagram to better illustrate study inclusion and exclusion processes

2. If mortality rates are low, compare them with international studies and discuss potential factors contributing to these rates in Iran – in discussion

3. Explain how deaths were attributed to colorectal cancer or competing risks, and describe the data collection method for death information.

4. Discuss the appropriateness of different outcomes like 1-year survival or 5-year survival to monitor colorectal cancer.

5. Specify if the study focuses on patient-related factors or healthcare service factors.

Discussion

1. Ensure alignment between methods, results, and discussion sections. The term "prognosis" should be introduced consistently throughout the manuscript.

2. Focus on variables included in the study and discuss their relevance.

3. Discuss whether the study findings are still applicable in the current context, considering healthcare advancements since the study period.

4. Align the discussion on socioeconomic status with the study's focus on care quality.

5. Clarify the collaborative program's strengths and its implementation.

6. Be consistent about sample size adequacy and its implications for the study's findings.

7. Clearly describe the study setting early in the manuscript for better contextual understanding.

8. Cross-reference and label figures appropriately.

Minor points:

Provide full spellings followed by abbreviations in parentheses.

Ensure consistent language and tone throughout the manuscript to avoid the appearance of sections written by different authors.

Reviewer #4: Firstly, I would like to thank the editor for inviting me to serve as a reviewer. In this study, the authors assess the differences in the survival of patients with colorectal cancer in two types of hospitals in Iran and draw conclusions about the quality of care. While the subject is interesting, there are some important conceptual issues that the authors must explain in their methodology before reaching any conclusions.

The title is confusing because it claims to investigate the quality of care in developing countries, whereas the study is conducted only on hospitals in Iran. It is incorrect to generalize the performance of Iranian hospitals to all developing countries. The abstract and title are misleading. The term "quality of care" is broad and can be used for various purposes. It would be better if the authors clarified that they are comparing the survival of colon and rectal cancer patients in two types of hospitals. It is questionable whether the differences between these two types of hospitals can serve as an indicator of quality of care, given that many hospital variables are ignored in this study.

The most important issue is that the authors must clearly state the names of the hospitals and the cities where they are located in their methodology, because there are a lot of variables which could effect the quality of care related to hospitals. One significant flaw in this study is that it includes only one NTNC and three TCC hospitals, yet attempts to generalize the performance of just four hospitals to all of Iran and developing countries. Furthermore, there is no clear definition of these hospitals in the text (e.g., high-volume?). Important variables about the hospitals, such as the number of physicians, nurses, and beds, are missing. Due to these flaws, I believe the article is not suitable for publication in its current format. The authors need to revise the title, abstract, and content of the study to more accurately reflect what they actually did. Additionally, they must provide detailed information about the hospitals included in the study.

Some minor issues:

- Note that the p-value cannot be equal to 0. Please change all instances of p-value = 0.00 to something like p-value < 0.001.

- Please mention in your methodology if there are any missing values and how you handled them.

- On pages 8 and 14, where you explain and use TMLE, please clarify which hospital type is considered as the reference in calculating the RR.

6. PLOS authors have the option to publish the peer review history of their article (what does this mean? ). If published, this will include your full peer review and any attached files.

**Do you want your identity to be public for this peer review?** For information about this choice, including consent withdrawal, please see our Privacy Policy .

Reviewer #1: **Yes: ** Mohamad Amin Pourhoseingholi

Reviewer #2: No

Reviewer #3: No

Reviewer #4: **Yes: ** Ali Golestani

---

## [Author Response · Author response to Decision Letter 1]

28 Nov 2024

Reviewer #1:

The study utilizes a robust methodology to assess survival disparities in non-metastatic colorectal cancer patients across different types of hospitals in Iran. The focus on high-volume teaching versus non-teaching hospitals provides valuable insights into the impact of hospital type on patient outcomes.

Comments:

1. Some parts of the manuscript are a bit confusing and could be better structured. For instance, the introduction should give a clearer and shorter summary of the research problem and goals via disparities.

Response: Thank you for your suggestion. We revised the introduction and clarified the research gap and goals.

2. The manuscript mentions the importance of the study, but it would be helpful to have a more in-depth discussion of other studies about colorectal cancer care differences in poorer countries. This would provide a stronger foundation for the research.

Response: As we addressed in the introduction, to our knowledge, no study has evaluated the quality of care in poorer countries so far.

3. It's great that you included different types of hospitals, but it would be good to explain in more detail why you chose these specific hospitals and how they represent the overall healthcare system.

Response: As an initial step, research studies like this have been voluntarily conducted by various centers to evaluate the quality of their care as part of routine practice, even years before a comprehensive framework for cancer care quality was developed. These initiatives represent an essential first step in assessing and enhancing the quality of cancer care, mainly when the appropriate organizations in the country have not yet established national systems and oversight mechanisms for such evaluations.

We focused on selecting hospitals prepared to collaborate and participate in this research. While our findings' results and internal validity are solid, we acknowledge that they may not be generalizable to the entire country. We have addressed this limitation in our discussion.

4. The information about the variables is thorough, but it would be useful to clarify exactly how you measured each variable and gathered the data. This would make it easier for others to repeat your study.

Response: We carefully addressed it in the "Variables" section.

5. The TMLE method is complex, so it would be helpful to explain it more clearly.

Response: We added a further description of the TMLE method. See lines: 231-242

6. Please explain why you chose the number of participants you did and if you calculated how many you needed to get reliable results (power analysis).

Response: We included all eligible patients admitted to the hospitals during the study period, ensuring that no cases were selectively excluded. While the sample size is adequate, we faced limited power in subgroup analyses, particularly rectal cancer cases in the NTNC hospital. However, the significant differences in patient survival suggest we had a sufficient minimum number of patients for these analyses. We acknowledge that a larger sample size would yield more robust results and recognize this as a limitation in our discussion.

7. The discussion does a good job of putting the results in context, but it could go deeper into why the differences in care exist.

Response: We enhanced the discussion in response to this comment, emphasizing how the differences between types of hospitals can lead to disparities in patient outcomes.

8. Based on your findings, please suggest specific actions or changes that could be made to improve the situation (or in further studies).

Response: We added more suggestions in the last part of the discussion and the conclusion sections. We believe that health policymakers and authorities should leverage the educational and scientific frameworks established in TCCS and provide support and resources to non-cancer facilities that may enhance the quality of cancer care across various healthcare settings. We also suggested scaling up this study by including additional centers to create comprehensive evidence regarding the quality of cancer care nationally and improve the national cancer control program.

Reviewer #2:

In the present study, the authors evaluated the outcomes and prognosis of colorectal, colon, and rectum cancers by the level of care provided in teaching and non-teaching non-cancer hospitals, using a data registry in Iran. The study subject is interesting. However, the use of language should be improved. Additionally, there are further comments and suggestions provided below:

**General comments**

1. There are some typos and grammatical issues within the text. Please proofread the entire manuscript.

Response: We proofread the entire manuscript and fixed the errors and typos.

2. Define abbreviations upon their first use in the text.

Response: We defined the abbreviations as suggested.

3. Use p<0.01 instead of p=0.00.

Response: Thanks. We corrected typos in presenting the P value.

**Abstract**

1. Please clarify and elaborate more on the background and objective of the study.

Response: We revised the background as suggested.

2. Check the number: RR=1.39, 95% CI: 1.01-2.022. Should it be 2.02 or 2.022? Revise it in the text if necessary.

Response: Thanks for noting this. We corrected the typo.

3. Revise the keywords based on MeSH.

Response: We updated the keywords based on MeSH terms.

**Introduction**

1. The SEER program in the US can also be discussed.

Response: We added results from the SEER program in the introduction.

2. In the second paragraph, please elaborate more and mention some of the findings from countries that have subnational or hospital-level data.

Response: We revised the introduction and incorporated the reviewer's suggestion in the new version.

**Methods**

1. Provide the exclusion criteria under the "Patients" subheading.

Response: We added the exclusion criteria as suggested. Please see lines: 148-153

2. Cite the relevant references for the guidelines used in the study (Variables subheading).

Response: We cited the relevant references for the guidelines used in the study.

3. It is recommended to provide an "Outcome" subheading and define the outcomes of interest for this study.

Response: We defined the outcome variables in the "Outcome" subheading line: 206-209

**Results and tables**

1. Table 1: Define what you mean by high, average, and low insurance coverage, and how you determine the risk of surgery. Also, it might be helpful to provide a column to represent the p-value between the groups for each variable.

Response: Thank you for the reminder; we have added the definitions of these two variables in the methods section under the "Variables" subheading. Regarding the p-value column in Table 1, we followed the STROBE guidelines, which recommend against reporting p-values for descriptive tables.

The statement in the STOBE guideline is:

"Inferential measures such as standard errors and confidence intervals should not be used to describe the variability of characteristics, and significance tests should be avoided in descriptive tables. Also, P values are not an appropriate criterion for selecting which confounders to adjust for in analysis; even small differences in a confounder that has a strong effect on the outcome can be important"

Ref: Vandenbroucke JP, von Elm E, Altman DG, Gøtzsche PC, Mulrow CD, Pocock SJ, Poole C, Schlesselman JJ, Egger M; STROBE Initiative. Strengthening the Reporting of Observational Studies in Epidemiology (STROBE): explanation and elaboration. Int J Surg. 2014 Dec;12(12):1500-24. doi: 10.1016/j.ijsu.2014.07.014. PMID: 25046751.

2. You mentioned that comorbidities were also considered. Please provide the baseline characteristics for comorbidities in Table 1 and clarify which comorbidities were considered.

Response: Thank you for your comment. It was a typo in our paper. In fact we used the "risk of surgery". Therefore, we replaced comorbidities with "risk of surgery".

Reviewer #3:

This study addresses a critical gap in understanding the disparities in care quality across different hospital types in the Eastern Mediterranean region. Below are my general comments and specific recommendations:

General comments:

1. The objective of the study, whether it focuses on the quality of care or prognosis, needs to be stated clearly at the outset.

Response: Thank you for your review and comments. While some steps and findings of this study mirror those of prognostic factor research, our primary focus is on evaluating care quality and treatment outcomes across various hospital groups.

We clarified this in the paper.

2. The rationale and importance of this study, as well as its contribution to the body of knowledge on colorectal cancer care quality, need to be more clearly articulated.

Response: We revised the introduction and addressed the concern raised by reviewer.

3. Ensure that the outcome measures and study setting are explicitly defined.

Response: We clarified the outcome definition in the methods section and devopted specific subheading for it.

4. Maintain consistent language and style throughout the manuscript.

Response: We proofread the entire manuscript.

5. Improve the writing style by starting with a stem statement followed by key points relevant to the study's objectives and direction.

Response: We revised the manuscript to enhance the style and ensure the consistency of concepts."

Specific sections:

Abstract: Please revise the abstract to reflect the recommendations provided for the entire article.

Response: We revised the abstract according to the suggestion.

** Introduction**

1. Include a section that describes the burden of colorectal cancer and the significance of monitoring and evaluating care outcomes.

Response: We added this information in the first paragraph of the introduction.

2. Justify the selection of input, process, and outcome indicators commonly used to monitor care quality, enabling readers to understand the global and regional context. This will help readers understand the international and regional context of the study.

Response: We made the necessary revisions and provided a comprehensive framework for the study, including the essential input, process, and output factors or indicators considered in the model. Figure 1 in the revised paper illustrates this framework clearly.

3. Follow statements such as "registry-based study has been advanced by a collaborative network" with an explanation of the network's function and contribution to improving care quality.

Response: We have carefully addressed it lines 110-114.

4. Describe the multidisciplinary team involved, the characteristics of well-equipped hospitals, and how these factors contribute to care quality.

Response: We added a description for the multidisciplinary team involved and other necessary information. (see lines 92-99).

5. Provide information on existing patient outcomes, survival rates, and process indicators used in colorectal cancer surgery. Discuss the advantages and disadvantages of these indicators.

Response: We addressed them in lines 72-81, respectively.

6. Include evidence that patient outcomes for colorectal cancer vary based on hospital type (teaching status, cancer specialization) and location.

Response: We added some evidence in the manuscript (please see lines 333-342).

7. Clearly describe the infrastructure gaps relevant to colorectal cancer care quality that the study aims to highlight.

Response: In this study we addressed the input, process and outome in care quality. In input, we considered some important factors in patient and tumor side, but presumed the hospital infrastructures are not significantly different in two types of hospitals as mentioned in lines 171-174.

8. Refine the objective statement using the SMART criteria (Specific, Measurable, Achievable, Relevant, Time-bound).

Response: Thanks for such a constructive comment. We revised the study objective accordingly.

** Methods**

1. Provide an overview of the healthcare systems in Iran, particularly colorectal cancer services, and any changes in service delivery over time.

Response: We added a parsgraph about colorectal cancer care in Iran (91-100).

2. Organize the Methods section to provide a clear contextual understanding of colorectal cancer treatment in Iran, followed by the role of different hospitals, the quality-network, and data collection processes. Conclude with detailed data management and analysis descriptions.

Response: We revised the methods section as recommended.

3. Clarify whether QRN-CRC is limited to specific hospitals or is a nationwide program. Provide the rationale for selecting the quality registry network and differentiate between hospital types.

Response: As an initial step, research studies like this have been voluntarily conducted by various centers to evaluate the quality of their care as part of routine practice, even years before a comprehensive framework for cancer care quality was developed. These initiatives represent an essential first step in assessing and enhancing the quality of cancer care, mainly when the appropriate organizations in the country have not yet established national systems and oversight mechanisms for such evaluations.

We focused on selecting hospitals prepared to collaborate and participate in this research. While our findings' results and internal validity are solid, we acknowledge that they may not be generalizable to the entire country. We have addressed this limitation in our discussion.

4. Explain how clinical registry data is collated, whether through hospital information systems or retrospective patient tracking and case note reviews.

Response: We provided this information and clarified the data collection procedure in a specific subheading named "Database" in the revised paper. Please check lines 155-167.

5. Specify the number of hospitals included in the study, the definition of high-volume hospitals, and the quality of data in the cancer registry.

Response: We provided details about the hospitals in the revised manuscript.

6. Summarize variables in a table with clear definitions to aid reader comprehension. Include guidelines for patient treatment based on cancer stages as appendices, if necessary.

Response: We completed the "variables" subheading and provided a figure for the framework of the variables used in this study.

7. Clearly define the study's outcome or dependent variables.

Response: we clarified the definition of the outcomes as the dependent variables at the end of the "variables" subheading.

8. Clarify the study duration and follow-up period. Provide details on patient follow-up timelines to help readers understand the duration of follow-up.

Response: We added the median of follow-up in this study. In addition, we provided a supplementary table and compared the censorship by different variables. (see Appendix Table 1)

9. Describe potential selection bias reasons and the steps taken to mitigate it.

Response: We compared the censored and non-censored patients ((see Appendix Table 1) and showed that there were no significant differences in each group according to different variables,

**Results**

1. Summarize the patient flow in a diagram to better illustrate study inclusion and exclusion processes

Response: Thank you for recognizing the importance of the "patient flow diagram" for this study and for your feedback. We added this diagram in a supplementary file.

2. If mortality rates are low, compare them with international studies and discuss potential factors contributing to these rates in Iran – in discussion

Response: Thank you for your comment, but we were not sure that this comparison can be addressed in this manuscript with different scopes and objectives of this paper. For an international comparison, we will need to measure net survival using the relative survival method. We provided such a comparison from 15 cancer subsites in another paper from our team:

Nemati S, Saeedi E, Lotfi F, Nahvijou A, Mohebbi E, Ravankhah Z, Rezaeianzadeh A, Yaghoobi-Ashrafi M, Pirnejad H, Golpazir A, Dolatkhah R, Alvand S, Ahmadi-Tabatabaei SV, Cheraghi M, Weiderpass E, Bray F,

---

## [Decision Letter · Decision Letter 1]

14 Jan 2025

PONE-D-23-43777R1Colorectal Cancer Care Quality in a Developing Country: Insights from a Comparison of Teaching and Non-teaching Hospitals in IranPLOS ONE

Dear Dr. Zendehdel,

Thank you for submitting your manuscript to PLOS ONE. After careful consideration, we feel that it has merit but does not fully meet PLOS ONE’s publication criteria as it currently stands. Therefore, we invite you to submit a revised version of the manuscript that addresses the points raised during the review process.

We look forward to receiving your revised manuscript.

Kind regards,

Sina Azadnajafabad, MD, MPH

Academic Editor

PLOS ONE

**Additional Editor Comments:**

Several major issues have been raised by the reviewers which need appropriate attention and revision of the manuscript accordingly.

Reviewers' comments:

Reviewer's Responses to Questions

**Comments to the Author**

1. If the authors have adequately addressed your comments raised in a previous round of review and you feel that this manuscript is now acceptable for publication, you may indicate that here to bypass the “Comments to the Author” section, enter your conflict of interest statement in the “Confidential to Editor” section, and submit your "Accept" recommendation.

Reviewer #1: All comments have been addressed

Reviewer #2: All comments have been addressed

Reviewer #3: (No Response)

2. Is the manuscript technically sound, and do the data support the conclusions?

Reviewer #1: Yes

Reviewer #2: Yes

Reviewer #3: Partly

3. Has the statistical analysis been performed appropriately and rigorously? 

Reviewer #1: Yes

Reviewer #2: Yes

Reviewer #3: No

4. Have the authors made all data underlying the findings in their manuscript fully available?

Reviewer #1: Yes

Reviewer #2: Yes

Reviewer #3: No

5. Is the manuscript presented in an intelligible fashion and written in standard English?

Reviewer #1: Yes

Reviewer #2: Yes

Reviewer #3: Yes

6. Review Comments to the Author

Reviewer #1: The revision is comprehensive. However, I suggest the authors to elaborate deeper into the implications of selection bias and unmeasured confounders as the potential limitation of study.

Reviewer #2: Thank you for implementing the comments. At this time, I have no additional comments or questions to raise.

Reviewer #3: The authors have attempted to address my concerns and feedback; however, critical issues remain regarding the focus and objectives of the stud and the added value it contributes to improving quality of care for colorectal cancer (CRC). While the authors conclude that care in teaching hospitals is better than in non-teaching hospitals based on hazard ratios, this finding alone does not sufficiently establish the broader importance or contribution of the study.

The rationale and importance of the study remain insufficiently addressed. The authors have not clearly articulated how their findings contribute to the body of evidence on CRC care quality. Specifically, the study does not explore what actionable insights can be drawn to improve care in non-teaching hospitals, nor does it contextualise findings within existing literature from similar healthcare systems or low- and middle-income countries.

- The focus on teaching status as the sole determinant of care quality oversimplifies a complex issue. It neglects to consider other critical factors such as healthcare system organisation, resource allocation, or regional disparities.

The authors cite the STROBE guidelines, which recommend avoiding p-values in tables describing variability of characteristics. However, the text includes a lengthy description of differences based on p-values. This inconsistency undermines the methodological clarity and should be addressed. The authors should clearly define the purpose of presenting p-values in the study and align their reporting with the cited guidelines.

The first objective—assessing quality of care independent of hospital type—is only partially addressed. The manuscript predominantly focuses on outcome measures (e.g., mortality and survival rates) without providing a comprehensive assessment of quality of care. Key dimensions such as safety, person-centred care, and accessibility are absent.

The assumption that hospital infrastructure is comparable is problematic. The authors acknowledge differences in infrastructure (e.g., advanced diagnostics, multidisciplinary teams in teaching hospitals) but do not explore how these differences affect quality. While adherence to guidelines is included, this alone does not represent a holistic evaluation of care quality.

The second objective—comparing high-volume hospitals and comprehensive cancer centres—requires further clarification and depth. I suggest that the authors explicitly address the following assumptions:

Quality of care is determined by hospital type. Please provide evidence from existing literature supporting the premise that teaching hospitals deliver better care due to multidisciplinary teams, structured adherence to guidelines, and access to advanced resources; and discuss why non-teaching hospitals, despite being high-volume, may lack comparable capabilities.

Hospital infrastructure and resources are comparable within categories. Clarify whether the included hospitals are truly representative of their respective categories. The authors assume that disparities are due to teaching status, yet they do not evaluate or account for unmeasured factors such as staffing, technology, or hospital policies.

Differences in quality can be measured by outcomes. Survival rates, guideline adherence, and process indicators (e.g., emergency surgeries) are valid proxies for quality of care, but the authors should acknowledge their limitations in fully capturing care quality dimensions.

Patient profiles are comparable after adjustment. Although adjustments were made for confounders, the possibility of bias remains, particularly due to differences in patient selection (e.g., more emergency cases in non-teaching hospitals). This limitation should be explicitly discussed.

Discussion and Contextualisation. The manuscript highlights disparities in outcomes between hospital types but does not explore the underlying causes or contextualise these disparities meaningfully. For instance: What specific processes or resources in teaching hospitals lead to better outcomes? How do these findings align with global evidence on CRC care disparities, particularly in similar healthcare systems?

I recommend that the authors expand the discussion to address these gaps and provide actionable recommendations for improving care in non-teaching hospitals.

Additional suggestions:

Provide more description/ data on hospital-specific characteristics, such as staffing levels, resource availability, and patient volumes, to strengthen the comparison.

Discuss the generalisability of findings, particularly given the small sample of hospitals included.

7. PLOS authors have the option to publish the peer review history of their article (what does this mean? ). If published, this will include your full peer review and any attached files.

**Do you want your identity to be public for this peer review?** For information about this choice, including consent withdrawal, please see our Privacy Policy .

Reviewer #1: **Yes: ** Mohamad Amin Pourhoseingholi

Reviewer #2: No

Reviewer #3: No

---

## [Author Response · Author response to Decision Letter 2]

28 Feb 2025

Reviewer #1: The revision is comprehensive. However, I suggest the authors to elaborate deeper into the implications of selection bias and unmeasured confounders as the potential limitation of study.

Response: Thanks for the comment. Please see lines: 401-408

2- Reviewer #3: The authors have attempted to address my concerns and feedback; however, critical issues remain regarding the focus and objectives of the study and the added value it contributes to improving quality of care for colorectal cancer (CRC). While the authors conclude that care in teaching hospitals is better than in non-teaching hospitals based on hazard ratios, this finding alone does not sufficiently establish the broader importance or contribution of the study.

Response: We have thoroughly revised the introduction and objectives of the manuscript and have added the necessary explanations.

3- The rationale and importance of the study remain insufficiently addressed. The authors have not clearly articulated how their findings contribute to the body of evidence on CRC care quality. Specifically, the study does not explore what actionable insights can be drawn to improve care in non-teaching hospitals, nor does it contextualise findings within existing literature

Response: Regarding the rationale and importance of the study we added more useful explanations in the introduction part of the manuscript. Additionally, the necessary explanations regarding which aspects of the healthcare system non-academic hospitals should enhance in the management of colon and rectal cancers to reduce the existing differences in patient survival have been added in the Discussion section, lines 338–345.

4- The focus on teaching status as the sole determinant of care quality oversimplifies a complex issue. It neglects to consider other critical factors such as healthcare system organisation, resource allocation, or regional disparities.

Response: In addition to what has been stated in the introduction of the manuscript, health system orientation, resource allocation, and regional disparities are important in evaluating cancer care quality. According to the methodology of healthcare quality assessment, factors such as structural aspects and resource allocation are considered as inputs. When process evaluation in healthcare is conducted using quality indicators and outcome assessment, identifying disparities and differences can justify further examination of inputs to determine the underlying causes of these variations (see lines 412-415). Naturally, incorporating all these factors into a single model within this study is not feasible, as it would complicate the understanding of causal relationships.

5- The authors cite the STROBE guidelines, which recommend avoiding p-values in tables describing variability of characteristics. However, the text includes a lengthy description of differences based on p-values. This inconsistency undermines the methodological clarity and should be addressed. The authors should clearly define the purpose of presenting p-values in the study and align their reporting with the cited guidelines.

Response: We revised tables and text and described differences based on confidence interval or p-value.

6- The first objective—assessing quality of care independent of hospital type—is only partially addressed. The manuscript predominantly focuses on outcome measures (e.g., mortality and survival rates) without providing a comprehensive assessment of quality of care. Key dimensions such as safety, person-centered care, and accessibility are absent.

Response: The “introduction” and “discussion” sections of the manuscript have been revised and we addressed our approach to assessment of cancer care quality and rationale of this study. According to the definition provided in the manuscript, the primary component of quality evaluated in this study is “effectiveness,” while other components—such as “acceptability,” “equity,” “optimality,” and “legitimacy”—are undoubtedly important but fall outside the scope of our study.

7- The assumption that hospital infrastructure is comparable is problematic. The authors acknowledge differences in infrastructure (e.g., advanced diagnostics, multidisciplinary teams in teaching hospitals) but do not explore how these differences affect quality. While adherence to guidelines is included, this alone does not represent a holistic evaluation of care quality.

Response: We have updated the Introduction section to thoroughly explain the study's rationale and approach. Initially, our quality assessment focuses on process indicators—such as adherence to clinical guidelines—and their impact on patient outcomes. Once we identify any outcome disparities, after controlling for potential confounders, we plan to delve deeper into the surgical department by evaluating its specific quality indicators. Furthermore, comparing hospital infrastructures may help reveal the underlying causes of these differences. Believing that structural factors influence process measures, which in turn affect outcomes, we designed our causal model to distinctly separate structural and process variables.

8- The second objective—comparing high-volume hospitals and comprehensive cancer centres—requires further clarification and depth. I suggest that the authors explicitly address the following assumptions:

Quality of care is determined by hospital type. Please provide evidence from existing literature supporting the premise that teaching hospitals deliver better care due to multidisciplinary teams, structured adherence to guidelines, and access to advanced resources; and discuss why non-teaching hospitals, despite being high-volume, may lack comparable capabilities.

Hospital infrastructure and resources are comparable within categories. Clarify whether the included hospitals are truly representative of their respective categories. The authors assume that disparities are due to teaching status, yet they do not evaluate or account for unmeasured factors such as staffing, technology, or hospital policies.

Response: Thank you for your suggestion. We have revised the "Introduction" section of the manuscript, incorporating a rational and historical approach in the literature review to ensure that all aspects of the suggested notes—regarding hospital categorization, infrastructure comparisons, and representativeness—are clearly addressed.

9- Differences in quality can be measured by outcomes. Survival rates, guideline adherence, and process indicators (e.g., emergency surgeries) are valid proxies for quality of care, but the authors should acknowledge their limitations in fully capturing care quality dimensions.

Response: We revised the introduction totally and also add a paragraph to the study limitation part of the “discussion” section (see lines 412-418).

10- Patient profiles are comparable after adjustment. Although adjustments were made for confounders, the possibility of bias remains, particularly due to differences in patient selection (e.g., more emergency cases in non-teaching hospitals). This limitation should be explicitly discussed.

Response: Thank you for this suggestion. In the limitations section of the Discussion, we explain that only the minimum essential dataset needed for a comprehensive comparison of hospitals was included (Please see lines: 406-408).

11- Discussion and Contextualisation. The manuscript highlights disparities in outcomes between hospital types but does not explore the underlying causes or contextualise these disparities meaningfully. For instance: What specific processes or resources in teaching hospitals lead to better outcomes? How do these findings align with global evidence on CRC care disparities, particularly in similar healthcare systems? I recommend that the authors expand the discussion to address these gaps and provide actionable recommendations for improving care in non-teaching hospitals. Provide more description/ data on hospital-specific characteristics, such as staffing levels, resource availability, and patient volumes, to strengthen the comparison.

Response: We have revised the Introduction section to comprehensively detail the study’s rationale and approach. As a first step in assessing quality of care, our focus is on process indicators—such as adherence to clinical guidelines—and their impact on patient outcomes. Once we identify any outcome disparities while controlling for potential confounders, we plan to further investigate the surgical department by examining its specific quality indicators. For instance, approved indicators related to the surgical process include achieving a dissection of at least 12 lymph nodes during colon cancer surgery and ensuring an interval of 8–12 weeks from the end of neoadjuvant chemoradiation therapy to the rectal cancer surgery. Additionally, comparing hospital infrastructures may help uncover the root causes of these differences. We have applied these strategies in separate studies. We added a paraghraph based on this pathway in the “discussion” section.

12- Discuss the generalisability of findings, particularly given the small sample of hospitals included.

Response: The proposed quality assessment model was developed based on a minimum essential dataset while considering the most influential and confounding factors affecting the outcomes of colorectal cancer surgery patients in two hospital groups. By evaluating the model’s feasibility and by encouraging the centers to implement the model and assess the outcomes derived from data collection, a disparity between the two hospital groups was ultimately observed, thereby enabling the generaliability of the model under similar conditions. It is important to note that the generalizability of findings similar to those in this study was not the primary objective.

---

## [Decision Letter · Decision Letter 2]

4 Apr 2025

PONE-D-23-43777R2Colorectal Cancer Care Quality in a Developing Country: Insights from a Comparison of Teaching and Non-teaching Hospitals in IranPLOS ONE

Dear Dr. Zendehdel, 

Thank you for submitting your manuscript to PLOS ONE. After careful consideration, we feel that it has merit but does not fully meet PLOS ONE’s publication criteria as it currently stands. Therefore, we invite you to submit a revised version of the manuscript that addresses the points raised during the review process.

We look forward to receiving your revised manuscript.

Kind regards,

Sina Azadnajafabad, MD, MPH

Academic Editor

PLOS ONE

Journal Requirements:

**Additional Editor Comments:**

Thanks for the previous revision. Please consider addressing the reviewers' comments on the presentation of the study objectives.

Reviewers' comments:

Reviewer's Responses to Questions

**Comments to the Author**

1. If the authors have adequately addressed your comments raised in a previous round of review and you feel that this manuscript is now acceptable for publication, you may indicate that here to bypass the “Comments to the Author” section, enter your conflict of interest statement in the “Confidential to Editor” section, and submit your "Accept" recommendation.

Reviewer #1: All comments have been addressed

Reviewer #3: All comments have been addressed

2. Is the manuscript technically sound, and do the data support the conclusions?

Reviewer #1: Yes

Reviewer #3: Yes

3. Has the statistical analysis been performed appropriately and rigorously? 

Reviewer #1: Yes

Reviewer #3: Yes

4. Have the authors made all data underlying the findings in their manuscript fully available?

Reviewer #1: Yes

Reviewer #3: Yes

5. Is the manuscript presented in an intelligible fashion and written in standard English?

Reviewer #1: Yes

Reviewer #3: Yes

6. Review Comments to the Author

Reviewer #1: (No Response)

Reviewer #3: The authors have addressed my earlier concerns. Nonetheless, the revised objectives remain broad and somewhat disjointed from the methods and results. I recommend refining the objectives to clearly focus on comparing survival outcomes and quality indicators between hospital types. Broader aims, such as registry development and quality model proposals, should be framed separately as background or future directions.

7. PLOS authors have the option to publish the peer review history of their article (what does this mean? ). If published, this will include your full peer review and any attached files.

**Do you want your identity to be public for this peer review?** For information about this choice, including consent withdrawal, please see our Privacy Policy .

Reviewer #1: **Yes: ** Amin Pourhoseingholi

Reviewer #3: No

---

## [Author Response · Author response to Decision Letter 3]

17 May 2025

6. Review Comments to the Author: the revised objectives remain broad and somewhat disjointed from the methods and results. I recommend refining the objectives to clearly focus on comparing survival outcomes and quality indicators between hospital types. Broader aims, such as registry development and quality model proposals, should be framed separately as background or future directions.

Response: Thank you for your comment. We have revised the objectives section to focus on the specific findings of the study.

---

## [Editor Report · Decision Letter 3]

5 Jun 2025

Colorectal Cancer Care Quality in a Developing Country: Insights from a Comparison of Teaching and Non-teaching Hospitals in Iran

PONE-D-23-43777R3

Dear Dr. Zendehdel,

We’re pleased to inform you that your manuscript has been judged scientifically suitable for publication and will be formally accepted for publication once it meets all outstanding technical requirements.

Kind regards,

Sina Azadnajafabad, MD, MPH

Academic Editor

PLOS ONE
---

## [Editor Report · Acceptance letter]

PONE-D-23-43777R3

PLOS ONE

Dear Dr. Zendehdel,

I'm pleased to inform you that your manuscript has been deemed suitable for publication in PLOS ONE. Congratulations! Your manuscript is now being handed over to our production team.

Kind regards,

on behalf of

Dr. Sina Azadnajafabad

Academic Editor

PLOS ONE